# Functional Training and Dual-Task Training Improve the Executive Function of Older Women

**DOI:** 10.3390/geriatrics8050083

**Published:** 2023-08-22

**Authors:** Alan Pantoja-Cardoso, Jose Carlos Aragão-Santos, Poliana de Jesus Santos, Ana Carolina Dos-Santos, Salviano Resende Silva, Newton Benites Carvalho Lima, Alan Bruno Silva Vasconcelos, Leonardo de Sousa Fortes, Marzo Edir Da Silva-Grigoletto

**Affiliations:** 1Postgraduate Program in Physical Education, Federal University of Sergipe, São Cristovao 49100-000, Brazil; dasilvame@gmail.com; 2Postgraduate Program in Health Sciences, Federal University of Sergipe, São Cristovao 49100-000, Brazil; prof.josecarlosaragao@gmail.com (J.C.A.-S.); newtonbenites@academico.ufs.br (N.B.C.L.); 3Postgraduate Program in Physiological Science, Federal University of Sergipe, São Cristovao 49100-000, Brazil; polianasantos.28@hotmail.com (P.d.J.S.); salvianoresende77@hotmail.com (S.R.S.);; 4Graduation in Physiotherapy, Federal University of Sergipe, São Cristovao 49100-000, Brazil; anacarolinaif94@gmail.com; 5Associate Graduate Program in Physical Education, Federal University of Paraiba, João Pessoa 58051-900, Brazil; leodesousafortes@hotmail.com

**Keywords:** aging, physical exercise, cognitive training, functional status, cognition

## Abstract

Functional training (FT) is a type of multicomponent training with emphasis on activities of daily living that stimulate different physical capacities in only one session. Dual-task training (DTT) is a type of training that simultaneously applies cognitive and motor stimuli. We investigated the effects of sixteen weeks of FT and DTT and eight weeks of detraining on older women’s inhibitory control, working memory, and cognitive flexibility. Sixty-two older women (66.9 ± 5.4 years; 27.7 ± 3.9 kg/m^2^) completed a 16-week intervention program comprising the FT (n = 31) and DTT (n = 31), and 43 returned after the detraining period. We used the Stroop Color Word Color test to evaluate inhibitory control, the Corsi Block Test to assess working memory, and the Trail Making Test to evaluate cognitive flexibility. Only DTT reduced the congruent response time between the pre-test and post-test (d= −0.64; *p* < 0.001), with no difference between the post-test and the detraining values (d = 1.13; *p* < 0.001). Both groups reduced the incongruent response time between the pre-test and post-test (FT: d = −0.61; *p* = 0.002; DTT: d= −0.59; *p* = 0.002) without a difference between groups. There were no significant differences in working memory and cognitive flexibility. Sixteen weeks of FT and DTT increased the inhibitory control of older women but not the working memory and cognitive flexibility, and these effects persisted after eight weeks of detraining.

## 1. Introduction

The main domains of executive function (EF) are inhibitory control, working memory, and cognitive flexibility [1]. Inhibitory control can prevent the allocation of focused attention on irrelevant or distracting information in internal or external environments, thus, allowing a person to keep focused on the task performed or relevant goal to be achieved [2]. Inhibitory control is essential for coping with the demands of daily life, such as avoiding distractions while walking or other daily activities [3]. Working memory constitutes the storage of information that will be manipulated and used to comprehend a task (e.g., listing, in order, a list of household objects or a market list for shopping) [4]. It is possible to note the importance of working memory for older people in language comprehension tasks and adherence to the use of medications [5,6]. Cognitive flexibility is an executive process that combines inhibitory control and working memory to adapt the course of thoughts or actions according to a situation’s changing demands without clear instructions [7]. Cognitive flexibility is fundamental to the mental health of older adults, which determines their ability to adapt to pressure situations or even daily threats related to conscious and unconscious behaviors. It is also known as the central point of an effective function in the control of internal states, controlling impulses to achieve higher goals [8].

Women are more susceptible to cognitive decline, especially after menopause [9,10]. This decline is observed in several cognitive domains, including working memory, attention, and processing speed for visual and auditory stimuli [10,11,12]. The reduction in cognitive abilities frequently indicates decreased EF [13]. EF encompasses a range of essential cognitive processes for concentration and attention. These processes allow for performing tasks automatically, quickly, and efficiently [1,14]. Thus, inhibitory control, working memory, and cognitive flexibility are important for older adults to perform daily activities. EF is associated with successful management of activities of daily living and social skills, contributing to health promotion, functional independence, autonomy, and improvement of the quality of life of older people [15,16].

Exercise effectively improves cognition and attenuates performance declines [17,18]. Adding cognitive tasks to training programs has shown significant benefits for different populations [19,20]. Multicomponent training has been particularly effective for the older population by stimulating brain areas such as the prefrontal cortex and dorsolateral cortex, besides promoting neurogenesis through increased brain-derived neurotrophic factor (BDNF) secretion by the skeletal muscle [18,21,22]. Despite the positive effects, multicomponent training is often applied analytically without approaching the specificity [23,24]. Thus, functional training (FT) is an alternative that explores the specificity of approaching exercises similar to the activities of daily living, with several positive effects to the older women population [25,26]. To the best of our knowledge, however, there have been no studies investigating the possible effects of FT on EF in older women.

Another option is the performance of two or more cognitive and motor activities or two different motor activities simultaneously, known as dual-task training (DTT) [27,28]. Gavelin et al. (2020) [29] investigated DTT protocols used in the literature and found significant effects on global cognition in older adults. Despite the considerable number of studies regarding this type of training, however, the comprehension of the effects of DTT on EF is challenging due to the heterogeneity among training protocols and the inconsistencies between studies [30]. This variation results from the limited methodological explanation provided in the scientific literature regarding DTT protocols.

Additionally, to the best of our knowledge, there is only one study examining the consequences of detraining on inhibitory control in older women [31]. Also, few studies compared DTT and single-task training protocols on all three main EF domains [30]. Therefore, we investigated the effects of sixteen weeks of DTT compared with FT and eight weeks of detraining on the inhibitory control, working memory, and cognitive flexibility of older women. We hypothesized that DTT would improve inhibitory control, working memory, and cognitive flexibility more efficiently than FT due to the training specificity principle. Furthermore, we assumed that the exercise effects on executive function would be maintained after eight weeks of detraining.

## 2. Materials and Methods

### 2.1. Study Design

The experimental study had repeated measures and parallel groups [32] lasting 32 weeks (June 2022 to February 2023). The first four weeks were used for initial measurements (pre-test) followed by 16 weeks for the training protocols, two weeks for the post-training measurements (post-test), eight weeks for detraining, and two weeks for the final measures (Figure 1). All tests and training procedures were conducted at the Department of Physical Education. Experienced professionals, blinded to the protocol performed by the participants, conducted all the tests. Also, one blinded researcher performed the data analysis. During the pre-test and post-test period, the participants were asked to not perform any type of physical exercise besides the testing procedures.

### 2.2. Participants

We recruited participants by disseminating posters and leaflets in the University surroundings using non-probabilistic recruitment methods (Figure 2). We randomized the participants with a 1/1 allocation rate based on their Stroop Color Word Test (SCWT) values, which were ordered in ascending order using Microsoft Excel software (Corp, Redmond, WA, USA). Participants were organized in ascending order based on the SCWT. From this, they were allocated into blocks and each assigned random values using Microsoft Excel software. Participants with the highest random value in each block were allocated to FT and those with lower values to DTT.

All participants fulfilled the following inclusion criteria: ≥60 years old; female; had no menstrual bleeding in the preceding 12 months; had no musculoskeletal or cardiovascular contraindications that would preclude exercise; had not undergone physical training for at least six months; had no color blindness, dyslexia, or severe mental or cognitive problems with medical report and achieved a minimum of 12 Montreal Cognitive Assessment scores [33,34]; and no previous experience in physical exercises. Afterward, all participants signed the informed consent form after all the procedures were explained. This study was approved by the Research Ethics Committee of the Federal University of Sergipe report no. 5.449.765 and registered in the Brazilian Registry of Clinical Trials under protocol RBR-2d56bt (available at https://ensaiosclinicos.gov.br/RBR-2d56bt (accessed on 1 November 2020).

### 2.3. Intervention Procedures

Professionals experienced in physical training for older people supervised the FT and DTT interventions. The professionals applied the protocols in different spaces within the Physical Education Department facilities of the university. There was at least one professional for every six participants to guarantee the participants’ safety and proper exercise executions.

The training intensity was monitored and adjusted based on the rate of perceived effort [35]. Both groups performed a total of 48 sessions with 3 weekly sessions from 42 to 72 h per session. During each FT session, five parts were completed: the first part included joint mobility and basic movements for preparation; the second part stimulated agility, speed, coordination, and muscle power; the third part explored muscle strength in basic patterns such as the squat, pull, push, and transfer; the fourth part stimulated the cardiorespiratory fitness using intermittent activities. Finally, breathing and stretching exercises were used as cool-down (Table 1; full table in Appendix A).

Every DTT session was divided into five parts: the first one focused on joint mobility and basic patterns combined with memory tasks; the second part used static and dynamic balance exercises combined with arithmetic and memory tasks; the third explored motor coordination exercises associated with reaction time and memory tasks; the fourth was based on group dynamics (coordinating activities with implement) with arithmetic and memory tasks; the fifth part was similar to the FT session with breathing and stretching exercises (Table 1; full table in Appendix A).

### 2.4. Procedures

#### 2.4.1. Anthropometric Characteristics

We assessed height using a portable stadiometer (Sanny, ES2030, São Paulo, Brazil). To measure body weight, we used a Lider P150C scale (São Paulo, Brazil) with a capacity of 200 kg and 100 g accuracy. The volunteer stood barefoot, wearing light clothes, with their heels together, and looking forward. We calculated the body mass index according to the protocols set by the World Health Organization’s calculation of weight (kg)/height (m^2^) [36].

#### 2.4.2. Executive Function

##### Inhibitory Control

The Stroop Color and Word Test (SCWT) is a neuropsychological test widely used for experimental and clinical purposes [34,37]. This test evaluates the ability to inhibit cognitive interferences, which occur when the processing of a stimulus’ characteristics affects another attribute’s simultaneous processing [35,38]. PsychoPy^®^ polarization version 2022 1.3 (https://pavlovia.org/ (accessed on 10 January 2022)) was used to construct the experiment’s stimuli and assembly.

The researchers applied the test on 15-inch screen computers using keyboards with yellow, blue, green, and red tapes attached to the letters “A”, “D”, “J”, and “L”, respectively. The participants were instructed to focus on the color of the word rather than the word itself. To familiarize the participants with the test, they initially performed 12 trials with feedback indicating whether their responses were correct or incorrect, along with relevant information about the test. Afterward, the participants completed 120 trials without receiving feedback.

These trials consisted of 60 congruent words (where the word’s meaning matched the ink color) and 60 incongruent words (where the word’s meaning and ink color diverged). The variables analyzed in the study were response time for congruent and incongruent trials, which showed good reliability for older women with an intraclass correlation coefficient of 0.92 and 0.91, respectively [39,40].

##### Memory Working

The Corsi Block Test (CBT) aims to assess visuospatial working memory, which retains, processes, and manipulates important information in the task context [37]. The software (PsychoPy^®^ version 2022 1.3) and devices used in the SCWT were used for the application. The test consisted of nine cubes distributed randomly on the screen; the participant selected them in the proposed order, progressing as she performed the task correctly [38].

Each cube was shown in an interval of 500 milliseconds for the task requirements. The participants were instructed to provide verbal responses and use gestures to indicate the sequence of cubes shown on the computer screen while the evaluators controlled the cursor. The composite score calculated by multiplying the sequencing (i.e., the number of levels achieved) by the number of correct responses was used to measure working memory. This score demonstrated an intraclass correlation coefficient greater than 0.80 for older people [39,40].

##### Cognitive Flexibility

The Trail Making Test (TMT) assesses cognitive flexibility and visual search strategies [41]. This test consists of two phases: A and B. In phase A (TMT-A), the participant should connect the numbers (i.e., 1 to 25) randomly distributed on the paper following the ascending order. The TMT-A test evaluates attention, visual scanning, speed, and fine motor coordination. In phase B (TMT-B), the participant should alternate between numbers and words sequentially (e.g., 1, A, 2, B, 3, and C until 25) to measure cognitive flexibility. In both phases, the participant was asked to perform the test as quickly as possible [42]. Phases A and B showed an intraclass correlation coefficient of 0.84 and 0.77, respectively [39].

#### 2.4.3. Sample Size

The sample size was calculated using the program G*Power version 3.1.9.7 (Erdfelder, Faul & Buchner, 1996, Kiel, Alemanha) [43], using the SCWT as the main outcome from the results of Coetsee & Terblanche (2017) [44]. Considering a repeated measures design with time × group interaction, we achieved a minimum of 44 older women (i.e., 22 participants per group) adopting a power of 80%, an alpha of 0.05, and an effect size of 0.69. Considering a possible sample loss of up to 20%, ten more participants were added, resulting in a total sample of 54 older women.

#### 2.4.4. Statistical Analysis

All the data were analyzed using the statistical software Jamovi 2.3.18.0 (The Jamovi project, 2022) [45]. The descriptive statistics were based on estimated marginal means, standard deviations, 95% confidence intervals for continuous variables, and absolute and relative frequencies for categorical variables. We performed a visual inspection of the variables to check the data distribution. Assuming normality (Shapiro–Wilk) and homogeneity (Levenne) of the variables, we compared the groups (i.e., FT and DTT) at the baseline using an independent t-test for continuous variables and the chi-square test for categorical variables.

Considering the experimental design, inferential analyses were performed using generalized linear mixed models, adopting groups and time (i.e., pre-test, post-test, and detraining) as fixed effects and participants’ values as a random effect to address individual variations in the repeated measures model. Based on that model, we investigated the interaction between time and group and the time and group effect separately. If a significant effect was observed (i.e., *p* < 0.05), pairwise comparisons were made using Bonferroni adjustment. Additionally, we calculated Cohen’s d for the main comparisons, interpreting the effect sizes as trivial (<0.2), small (0.2 to 0.49), moderate (0.50 to 0.79), and large (≥0.80) [46,47].

## 3. Results

After 16 weeks of training, 86% (62) of the participants performed the post-test measurements and no adverse effects or accidents were reported. After the detraining phase, 43 participants remained and completed the evaluations, with 69.4% from the FT group and 50% of the DTT group. Figure 3 presents the withdrawals that occurred throughout the protocol. Table 2 provides an overview of the participants’ anthropometric, sociodemographic, cognitive level, and medical history characteristics at the baseline, indicating no significant differences between the groups.

In the SCWT, we observed group*time interaction (χ2 (4) = 7.77; *p* = 0.021), time (χ2 (2) = 23.51; *p* < 0.001), and group (χ2 (2) = 4.23; *p* = 0.04) effects in the congruent response time. Specifically, we detected a small non-significant reduction when comparing pre-test versus post-test in the FT (d = −0.29; *p* = 1.000), while the reduction was trivial and non-significant between post-test and detraining (d = −0.11; *p* = 1.000). In the DTT group, we found a moderate significant reduction in pre-test versus post-test (d = −0.64; *p* < 0.001) and a small non-significant reduction in post-test versus detraining (d = −0.45; *p* = 0.966) (Figure 3A). The DTT group showed a larger decrease across times.

Still regarding the SCWT, considering the incongruent response time, we detected a time (χ2 (2) = 23.29; *p* < 0.001) without a group (χ2 (2) = 0.163; *p* = 0.687) and group*time interaction (χ2(4) = 2.77; *p* = 0.250) effects. Between the time points, we detected a moderate reduction in the pre-test versus post-test in FT (d = −0.61; *p* = 0.002) and a trivial decrease in the post-test versus detraining (d = −0.11; *p* = 1.000). In the DTT, we found a moderate reduction in the pre-test versus post-test (d = −0.59; *p* = 0.002) and a small reduction in the post-test versus detraining (d = −0.24; *p* = 1.000) (Figure 3B).

In the CBT, considering the composite score, there was a group (χ2 (2) = 3.422; *p* = 0.03) without time (χ2 (2) = 3.596; *p* < 0.166) and group*time interaction (χ2 (4) = 1.202; *p* = 0.548) effects. The effect size of FT versus DTT was moderate at the pre-test (d = 0.55; *p* = 0.02), trivial at the post-test (d = 0.18; *p* = 1.000), and small at the detraining (d = 0.23; *p* = 1.000) (Figure 3C).

Regarding TMT-A, there was a group effect (χ2 (2) = 8.33; *p* = 0.004) but no time effect (χ2 (2) = 3.67; *p* = 0.159) or group*time interaction (χ2 (4) = 1.63; *p* = 0.443) effects. The FT versus DTT showed a small effect size at the pre-test (d = 0.37; *p* = 0.122), a trivial effect size at the post-test (d = 0.18; *p* = 0.005), and a large effect size at the detraining (d = 0.90; *p* = 0.003) (Figure 3D).

In TMT-B, there was a group effect (χ2 (2) = 47.198; *p* < 0.001) but no time effect (χ2 (2) = 5.853; *p* = 0.054) or group*time interaction (χ2 (4) = 0.052; *p* = 0.974). The FT versus DTT showed a medium effect size at the pre-test (d = 0.76; *p* < 0.001), while at the post-test and the detraining they showed a large effect size (d = 0.84; *p* < 0.001; d = 0.94; *p* < 0.001) (Figure 3E).

In the TMT B–A difference, there was a group effect (χ2 (2) = 42.298; *p* < 0.001) but no time effect (χ2 (2) = 5.121; *p* = 0.077) or group*time interaction (χ2 (4) = 0.510; *p* = 0.775). The FT versus DTT at the pre-test, post-test, and detraining showed large effect sizes (d = 1.05, *p* = 0.001; d = 0.90, *p* = 0.005; d = 1.02, *p* = 0.006, respectively) (Figure 3F).

## 4. Discussion

To our best knowledge, this is the first study to demonstrate that FT attenuates the decline in inhibitory control and keeps the adaptations even after eight weeks of detraining of older woman. Our main finding was that the FT and DTT protocols reduce incongruent response time after 16 weeks and their effects last for up to eight weeks of detraining. Furthermore, we observed that both protocols did not change in working memory and cognitive flexibility. Thus, we did not confirm our initial hypothesis that DTT would be superior to FT but we found an attenuation of the decline in the executive function of the DTT after eight weeks of detraining.

We observed that both DTT and FT decreased the incongruent response time similarly. Martinez-Navarro et al. (2021) [22], on the other hand, pointed out that the association between physical and cognitive training showed superior results compared with each training alone. In our study, both training protocols possibly stimulated different aspects of executive function. The DTT, however, aimed to stimulate the specific domains of executive function in each block, whereas FT explored training specificity using different approaches to increase the exercise intensity, possibly stimulating brain areas such as the prefrontal cortex and anterior cingulate cortex that are related to positive adaptations in executive function [48,49].

Although both groups reduced incongruent response time, the DTT group maintained this variable after the detraining, unlike the FT group. We stated that the effects lasted up to 8 weeks since there was no difference between the post-test and post-detraining measurement, indicating that the values between these points were not different, and the effect size was small (d = −0.24), thus reinforcing this statement. Our findings corroborate those of Blasco-Lafarga et al. (2020) [31], who found an attenuation of the decline in inhibitory control after 14 weeks of detraining in older women that practiced multicomponent physical–cognitive training programs. Thus, adding a second cognitive task would possibly increase the cognitive reserve and maintain the executive function after detraining periods. We did not find any effect on the working memory between the groups. Based on the effect sizes examination, however, the DTT group showed a small effect while the FT group produced a trivial effect. Chainay et al. (2021) [50] showed that neither the cognitive or physical isolated group nor even cognitive and physical stimuli together demonstrated superior working memory performance. Jardim et al. (2021) [51] investigated DTT at moderate intensity compared to an inactive control group and observed an increase in immediate memory only in the DTT group.

These results reveal inconsistencies in the literature regarding the actual impact of exercise interventions on working memory and the scarcity of studies investigating this variable [52]. Wu et al. (2023) [30] showed in a meta-analysis that single- and dual-task interventions had equal efficacy on working memory compared to non-intervention groups, supporting our findings regarding visuospatial working memory.

Brown et al. (2009) [53], in addition, compared the effects of the balance and strength exercise interventions versus the flexibility and relaxation group on the forward digit extension test to measure working memory in older people (19 men and 135 women). The effect size of the balance and strength intervention was trivial (d = 0.05). In contrast, flexibility and relaxation intervention reduced the performance, with a small effect size (d = 0.21), indicating that the balance and strength intervention at least maintained the working memory performance. Here, we used a training method similar to the balance and strength exercise intervention and found a small effect size increase (d = 0.44), higher than that of Brown et al. (2009) [53]. Our larger effect size was likely due to the cognitive tasks inserted in the DTT that stimulated the working memory subfunction.

In TT-A and -B, we did not observe any significant changes. Park (2022) [28] compared DTT to balance interventions and found a reduced execution time on the TMT-B that measures cognitive flexibility. Park (2022) [28] found a small effect size for DTT (d = 0.24) and a trivial effect for balance interventions (d = 0.09), indicating a more impactful effect of the DTT on cognitive flexibility. In our study, the DTT showed a negligible effect size (d = 0.16), smaller than the one observed by Park (2022) [28], which might be due to the lengthy duration of our intervention. Park (2022) [28] applied a four-week intervention while we applied 16 weeks, perhaps indicating an increased potential for learning with the proximity between measurements.

In a group of healthy older adults, cognitive flexibility increased after six weeks of high-intensity interval training to a greater extent than in those who participated in the continuous group of moderate intensity and resistance training [54]. Regarding the effects of functional exercises, our study corroborates that of Forte et al. (2013) [54,55], who did not detect a decrease in the execution time in the TMT-B, used to measure cognitive flexibility after 16 weeks of intervention [55]. Even without the increase in cognitive flexibility, maintaining performance in this variable is an important factor for the quality of life of this public [8].

In this context, the present study reinforces the current literature and introduces a novel approach that professionals can use to improve the inhibitory control of older women. Moreover, we add information concerning the consequences of detraining; additional research is warranted in this regard, encompassing longer detraining periods, larger groups, and physiological measures linked to the executive function.

It must be noted that both DTT and FT did not enhance the cognitive flexibility of older women. However, we could possibly find different results after an extended intervention with specific stimulus for this executive function domain. Nonetheless, the lack of a control group is a concern since it is unknown whether the observed changes happened randomly, particularly during the detraining period. Despite this limitation, our findings suggest that DTT and FT interventions could benefit seniors.

A limitation of the present study was the absence of tests related to physical fitness, such as aerobic fitness or muscle strength. However, other studies using similar training methods such as that of Aragão-Santos et al. [56], Resende-Neto et al. [26], Resende-Neto et al. [57], Brustio et al. [58], and Martinez-Navarro [22] are consistent regarding the increase in physical fitness through the application of FT and DTT. In addition, Weinstein et al. [59] and Predovam et al. [60] found a connection between aerobic fitness and executive function in older participants. This leads us to believe that the training protocols we used are effective for improving aerobic fitness since there was an increase in EF. This observed improvement in EF provides support for this statement. Nevertheless, further investigations are required to establish whether this association applies to other physical capabilities besides aerobic capacity. Second, we did not analyze physiological measures, such as heart rate and blood pressure, which may provide further support for the findings. Third, our study was conducted with older women of varying educational status, indicating heterogeneity. Despite these limitations, we believe that the results of our study are important and can contribute to the field of research.

Based on our results and limitations, it is worthwhile to research further the effects of DTT and FT on EF by exploring evaluation methods more similar to real-life scenarios, such as the timed-up and go test or the walking speed test combined with a second task, which could be a motor or cognitive task [22,26,56,57,58]. Another point is to investigate the relationship of different EF tests with physical fitness measures in older people. Besides, future research should approach men and women to examine sex differences in EF and response to physical exercise. Finally, developing more DTT protocols to use this type of training in different contexts is essential.

## 5. Conclusions

Sixteen weeks of FT and DTT effectively increased inhibitory control; however, only the DTT maintained this effect after eight weeks of detraining in older women. Both training programs, however, could not enhance working memory and cognitive flexibility. Nevertheless, exercise protocols are suitable for older women, showing safety and efficacy with improvement of at least one domain of executive function.

## Figures and Tables

**Figure 1 geriatrics-08-00083-f001:**
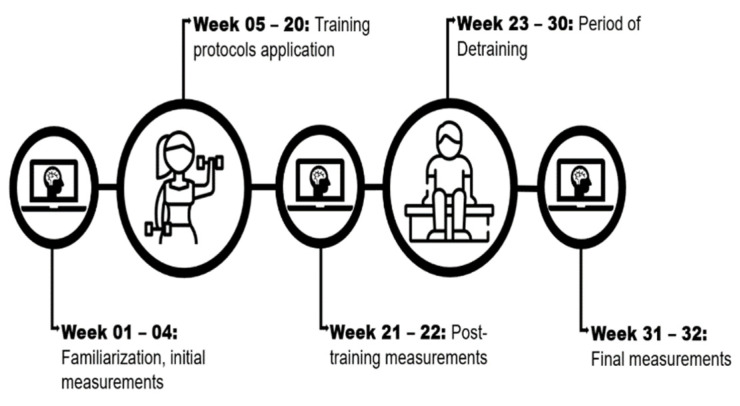
Experimental design.

**Figure 2 geriatrics-08-00083-f002:**
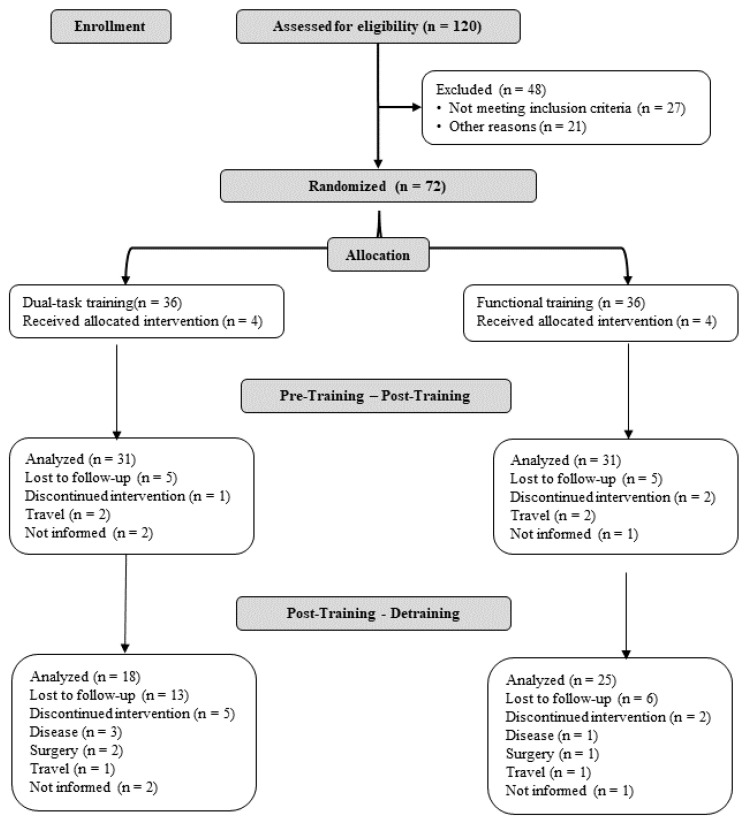
Flowchart of the screening and allocation of the participants.

**Figure 3 geriatrics-08-00083-f003:**
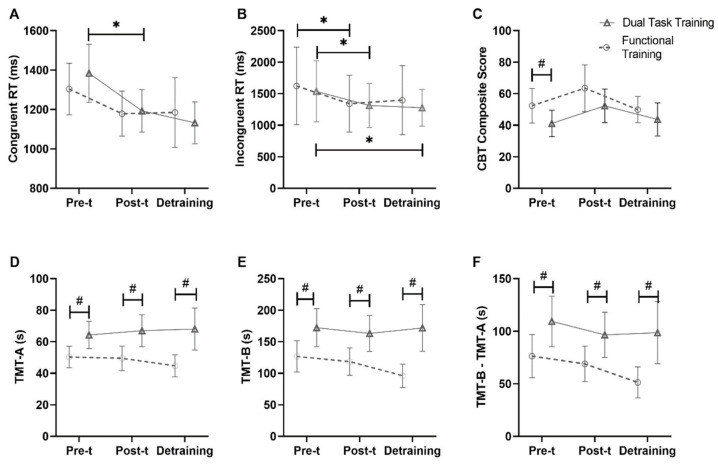
Effects of interventions and detraining period on executive function variables in older women. Specifically, Congruent RT (**A**) and Incongruent RT (**B**) for the inhibitory control, CBT composite score (**C**) for the working memory, and TMT-A (**D**), TMT-B (**E**), and TMT-B–TMT-A (**F**) for the cognitive flexibility. Values showed as means ± 95% confidence intervals (CI). Effects are derived from generalized mixed models using Gamma distribution. * Indicates between-times differences; # indicates between-groups differences for the same time point. CBT: Corsi Block Test; RT: response time. Pre-t: Pre-training; Post-t: Post-training.

**Table 1 geriatrics-08-00083-t001:** Functional training and dual-task training Program description.

	1st Part	2nd Part	3rd Part	4th Part	5th Part
FT	Frontal run skipping in a walking pattern	Latter agility	Deadlift	Relay	Single stretching + breathing
		Battle rope	Goblet squat		
	High-knee skips	Med ball throw	Farmer’s walk		
		Climbing on and off the step	Rowing		
		Agility between cones	Pushing		
DTT	Standing Stretch Patterns + Mobility and Stability + Evocation	Bipodal balance with feet together, one glued to the other + Balancing a ball, throwing from one hand to the other in front	Stationary march + Progression of steps from 2 to 2 and from 3 to 3 according to the command	Pass the ball to the side (calling for colors) in an isometric squat position, counting the number of turns	Single stretching + breathing
		Unipodal balance with assisting stick, in pairs, evoking the name of a class without repeating what has already been said	Lateral displacement with side shift		
		Straight line walking with feet on edge of center strip balancing baton			

Note: DTT = dual-task training; FT: functional training.

**Table 2 geriatrics-08-00083-t002:** Descriptive anthropometric, sociodemographic, cognitive level, and medical history data of the participants at the baseline.

	DTT (n = 31)		FT (n = 31)		Total (n = 62)		
Variables	Mean ± SD	CI 95%	Mean ± SD	CI 95%	Mean ± SD	CI 95%	*p*-Value
Age (years)	67 ± 5	65–69	66 ± 5	64–68	66 ± 5	65–68	0.405
Body Mass (kg)	66.3 ± 11.2	62.1–70.4	65.4 ± 8.9	62.1–68.7	65.8 ± 10.1	63.3–68.4	0.745
Height (m)	1.53 ± 0.05	1.51–1.55	1.55 ± 0.05	1.53–1.57	1.54 ± 0.05	1.53–1.54	0.112
BMI (kg/m^2^)	28.3 ± 4.3	26.7–29.9	27.1 ± 3.5	25.8–28.4	27.7 ± 3.9	26.7–28.7	0.25
MoCA (score)	21.0 ± 4.3	19.4–22.6	22.3 ± 3.7	20.9–23.7	21.6 ± 4.1	20.6–22.7	0.208
Education (relative and absolute frequency)							
Incomplete Elementary	22.6 (14)		14.5 (9)		37.1 (23)		0.694
Complete Elementary	3.2 (2)		3.2 (2)		6.5 (4)		
Incomplete High School	1.6 (1)		1.6 (1)		3.2 (2)		
Complete High School	19.4 (12)		22.6 (14)		41.9 (26)		
Incomplete Undergraduete degree	1.6 (1)		1.6 (1)		3.2 (2)		
Complete Undergraduete degree	1.6 (1)		6.4 (4)		8.5 (5)		
Medical History (relative and absolute frequency)							
Hypertension	59.5 (22)		40.5 (15)		59.7 (37)		0.07
Depression	80.0 (4)		20.0 (1)		8.1 (5)		0.162
Diabetes	60.0 (9)		40.0 (6)		24.2 (15)		0.374
Body pain	51.2 (22)		48.8 (21)		70.5 (43)		0.632

Note: BMI = body mass index; SD = standard deviation; CI = confidence interval; MoCA = Montreal Cognitive Assessment; DTT = dual-task training; FT = functional training.

## Data Availability

Data are available upon request.

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
