# Peer review of "Functional Training and Dual-Task Training Improve the Executive Function of Older Women"

_geriatrics, 2023, doi:10.3390/geriatrics8050083_

Round 1
Reviewer 1 Report
Basic Reporting
The manuscript reports a result of the effects of 16 weeks of functional training and dual-task training (and 8 weeks of detraining) on inhibitory control, working memory, and cognitive flexibility in older women. The results showed that only DTT reduced the congruent response time between the pre-test versus post-test, with no difference between the post-test and the detraining values. In addition, both groups reduced the incongruent response time between the pre-test and post-test without differences between groups. Although the manuscript is a little difficult to follow, the authors have done a decent job of providing an informative and meaningful addition to the current study field.
However, there are several changes that the authors are encouraged to revise to elevate the overall contribution of the paper to this research field.
Abstract:
Line 13: Please define FT and DTT for the reader
Line 15: superscript 2
Introduction:
In my opinion there are way too many paragraphs, and this doesn't help the flow. Perhaps condensing your main thoughts to 4 or 5 paragraphs (see below) would be best to ensure flow for the reader. I think the flow would be best if your thoughts/ideas were arranged like so - importance of cognitive function/flexibility (especially noting EF), cognitive decline in women, how we can attenuate or improve cognitive function (exercise - describe some studies here), DTT and details, and end with your aims/hypotheses.
Methods:
Were participants exercising during the pre-testing 4 weeks?
Can't see figure 1 at all
Were the participants allowed to exercise or train at all during the two-week post-testing time? This would alter the eventual outcome as if participants were not able to exercise/train then would the detraining technically be 10 weeks?
Can you further explain lines 91-93? I don't understand the high and low score groups?
Did you perform any test to measure baseline aerobic fitness?
Were participants left or right-handed?
Intensity measured by RPE is probably not the best way to monitor intensity or ensure that the exercise is delivering the desired physiologic effect.
Would tables 1 & 2 be better suited to a supplementary materials section?
Line 139-140: What are the WHO protocols for calculating BMI? It seems as though it's a simple calculation of weight (kg) / height (m2)
Line 159: needs a period at the end of the sentence
Line 188: extra space between "...power of" and "80%"
Line 208: extra ")"
Results:
Line 210: what does "86% of the participants performed the measurements" mean? Do you mean post-testing measurements? Please clarify
If there is no measure of aerobic capacity pre and post, how do we even know that the exercise intervention worked?
If table 3 is results of baseline demographics, would it be in the results or methods section?
Discussion
Figure 3B - shows statistical significance in the DTT group from pre to detraining but in the text you have p = 1.000? (line 231). Further, you mention in the discussion that their effects lasted up to eight weeks of detraining but these results were not significant? Please clarify
Line 267: you have FE? You also have this at other places throughout the manuscript, what is this? Please clarify and define if necessary
Line 274: should you have a reference year and number? (same on lines 281, 283, 287 and maybe more?)
Interesting that 16 weeks of exercise had no effect on TMT-A and TMT-B
Are there any limitations to the current study?
Line 322: why is there a number "5" present? Or is that part of 5. Conclusions? Please clarify and edit
Line 323: increased or improved?
Any future research recommendations? The limitations and conclusions need some bulking up in my opinion
Other small observations
Why are there numbers by the keywords?
Why are the authors affiliations not in numerical order?
IRB statement / consent form statement / data availability statement are all incorrect in my opinion as this was a human subject study and you did generate and record data?
I think you need to revise the entire manuscript for grammatical errors. If there is not one on the paper, it may be a good idea to get a native English speaker to do this. Overall, I think moderate editing is needed in this case.
Reviewer 2 Report
Minor revision required :
1. Table 3 is inappropriate. 'Dots' are replaced by 'comma'. P-values in the same table need to be corrected.
2. Though the authors have written in the methodology that informed consent was taken, they wrote 'not applicable' in the consent statement. This needs to be addressed.
3. Data Sharing statement should be made clear if the data will be provided on request.
The language can be improved, making it easy to understand.
Reviewer 3 Report
Dear Authors,
Overall the manuscript is well written but some issues need to be solved before its final publication.
It's not clear to me how the studies were carried out in time. The description of the course of the study doesn't correlate to figure 1.
Materials and Methods
2.1.Study design
This experimental study had repeated measures and parallel groups [25] lasting 33 weeks (June 2022 to February 2023). The first four weeks were used for initial measurements (pre-test), followed by 16 weeks for the training protocols, two weeks for the post-training measurements (post-test), eight weeks for detraining, and two weeks for the final measures (Figure 1).
4 + 16 + 2 + 8 + 2 = 32 ????
Figure 1 shows the Experimental design of the study in 30 weeks.
The first four weeks were used for initial measurements (pre-test) (1-4 weeks), 5-20 weeks followed by 16 weeks for the training protocols, 21 -22 weeks the post-training measurements (post-test), 23 – 30 eight weeks for detraining, 31 -32 two weeks for the final measures???
Line 51
Bad abbreviation used.
However, to the best of our knowledge, no studies have been conducted about the possible effects of FT on FE (EF???) in older women.
Line 56 -58
Bad abbreviation used
However, despite the considerable number of studies about DTT, the comprehension of the effects of DTT on FE (EF???) becomes more challenging due to the heterogenity among protocols and the inconsistencies between studies [21].
Line 63
Bad abbreviation used
Also, few studies compare DTT and single-task training protocols on all three main FE (EF???) domains [21].
Round 2
Reviewer 1 Report
Thank you for your resubmission.
In my opinion, I still believe the following 3 points need to be addressed:
- lack of aerobic fitness testing pre and post to show that the exercise intervention actually worked
- thorough future research ideas/thoughts and conclusions
- concise tables and figures
Nice job in improving the quality of English.
